# Continuous Positive Airway Pressure Treatment in Patients with Alzheimer’s Disease: A Systematic Review

**DOI:** 10.3390/jcm9010181

**Published:** 2020-01-09

**Authors:** Veronica Perez-Cabezas, Carmen Ruiz-Molinero, Jose Jesus Jimenez-Rejano, Gloria Gonzalez-Medina, Alejandro Galan-Mercant, Rocio Martin-Valero

**Affiliations:** 1Department of Nursing and Physiotherapy, University of Cadiz, 11009 Cadiz, Spain; veronica.perezcabezas@uca.es (V.P.-C.); carmen.ruizmolinero@uca.es (C.R.-M.); gloriagonzalez.medina@uca.es (G.G.-M.); alejandro.galan@gm.uca.es (A.G.-M.); 2Department of Physiotherapy, University of Seville, 41009 Seville, Spain; 3Department of Physiotherapy, University of Malaga, 29071 Malaga, Spain; rovalemas@uma.es

**Keywords:** continuous positive airway pressure, Alzheimer disease, neurocognitive function

## Abstract

Background: Epidemiological studies have suggested a pathophysiological relationship between obstructive sleep apnea syndrome (OSAS) and Alzheimer’s disease (AD). The aim of this study is to evaluate the treatment of obstructive sleep apnea with continuous positive airway pressure (CPAP) in AD and its relationship with neurocognitive function improvement. Methods: Systematic review conducted following PRISMA’s statements. Relevant studies were searched in MEDLINE, PEDro, SCOPUS, PsycINFO, Web of Science, CINAHL and SportDicus. Original studies in which CPAP treatment was developel in AD patients have been included. Results: 5 studies, 3 RCTs (Randomized controlled trials) and 2 pilot studies. In all RCTs the CPAP intervention was six weeks; 3 weeks of therapeutic CPAP vs. 3 weeks placebo CPAP (pCPAP) followed by 3 weeks tCPAP in patients with AD and OSA. The two pilot studies conducted a follow-up in which the impact on cognitive impairment was measured. Conclusions: CPAP treatment in AD patients decreases excessive daytime sleepiness and improves sleep quality. There are indications that cognitive deterioration function measured with the Mini Mental Scale decreases or evolves to a lesser extent in Alzheimer’s patients treated with CPAP. Caregivers observe stabilization in disease progression with integration of CPAP. More research is needed on the topic presented.

## 1. Introduction

Obstructive sleep apnea (OSA), the most common type of sleep apnea, consists of repetitive episodes of upper airway obstruction occurring during sleep [1]. Severe OSA, in addition to sleep fragmentation, produces intermittent hypoxemia and significant oxidative stress [2]. OSA represents a very important risk factor for stroke, hypertension, cardiovascular disease, hyperlipidemia, atrial fibrillation, type 2 diabetes mellitus and promote mechanisms that can increase a risk of dementia [3].

Epidemiological studies have suggested a pathophysiological relationship between obstructive sleep apnea syndrome (OSAS) and Alzheimer’s disease (AD) [4,5]. Several research works have suggested that OSAS patients were involved in short-term and working memory deficit, which was probably correlated with hypoxia-related change in hippocampal deterioration [6]. Dementia, a syndrome caused primarily by AD and cerebrovascular disease, is the process that leads to increased disability, dependence and use of social and health resources in the elderly [7]. Continuous positive airway pressure (CPAP) treatment in OSAS patients implies a partial improvement in their cognitive dysfunction [8] and may also facilitate neural rehabilitation. Therefore, in the early stages of AD, CPAP may delay disease progression. For this reason, we emphasize the importance of OSAS detection and timely intervention in these patients. Health professionals should identify OSAS problems in the AD early stages. Alzheimer’s is the process that leads to greater disability, dependence and use of socio-health resources in the elderly. Epidemiological studies of the projected prevalence of dementia estimate that the number of people with dementia in the world will double every 20 years, reaching 81.1 million in 2040 [7].

The main aim of this study is to detect the level of evidence and grades of recommendation regarding therapeutic respiratory interventions of CPAP in patients with AD. In addition, it aims to identify the impact of the intervention with CPAP on cognitive function and disease progression.

## 2. Material and Methods

### 2.1. Study Design

A systematic review literature search was carried out to identify all possible studies that might help to answer the research aim. This review was conducted following PRISMA’s statements [9]. The review protocol is available in PROSPERO (CRD42017078517). The PRISMA checklist is attached appendix 1. In order to approach the topic widely, a bibliographic search was made in the following electronic databases: PubMed, PEDro, SCOPUS, SPORTDiscus, CINHAL and Web of Science. A search was also made in the Health Sciences Library, but no additional resources were obtained.

### 2.2. Search Strategy

The keywords used were: “continuous positive airway pressure”, “Alzheimer”, “CPAP”, “physical therapy modalities” and “neurocognitive function”. All of them were used with the boolean operators AND and OR, suggesting the database in question. Given the specificity of the topic, the search was often 0. The review was updated from February to July in 2019. The following are only the positive results.

In Pubmed the search was undertaken through Mesh, identifying the terms “Alzheimer” AND “continuous positive airway pressure” obtaining a total of 13 documents. By filtering “clinical trial” the search resulted in a total of 5 matches.

For the review in SPORTDiscus the terms “Alzheimer” AND “CPAP or continuous positive airway pressure” were introduced without result. By combining “alzheimer” with “neurocognitive function” we also did not find any documents.

In PEDro we use the advanced search; in the section “abstract and title” we wrote the term “alzheimer” and in “therapy” we selected “respiratory therapy”. The result was only 1 document.

The combination of all proposed terms was attempted at the Cochrane Library. Results were only obtained by introducing “Alzheimer” and “CPAP”. In the space of “reviews” there was 1 document and “trials” there were 9.

In CINHAL the search was positive with “Alzheimer AND CPAP or continuous positive airway pressure”, with a total of 19 documents. With the filter of “full text” we were left with 9 and when filtering with “academic journals” with 6 documents.

The SCOPUS search was attempted with all terms. Finally, we only included “alzheimer AND” continuous positive airway pressure with a result of 29 documents. The filter we used was document type (articles and reviews), limiting the results to 25 documents.

In the Web of Science we used the main collection with the words “alzheimer” AND “continuous positive airway pressure”, with a result of 29. If we associated any other term, the result was 0. With the “review” and “article filter” we were left with 18 documents.

After eliminating duplicates, 12 documents were obtained. When reading the title and abstract 5 were excluded, because they had no relation with the topic. When accessing the complete text, one more was excluded, because it was written in Chinese. The entire process is described in Figure 1.

### 2.3. Inclusion and Exclusion Criteria

Inclusion criteria were constructed using the PICO (population, intervention, control/comparison and outcomes) model. First, the population included samples of people diagnosed with Alzheimer’s disease. In the studies reviewed, the diagnostic criteria of the National Institute of Neurological and Communicative Disorders and Stroke and the Alzheimer’s Disease and Related Disorders Association (NINCDS-ADRDA) were used [10]. For OSA polysomnography determines apnea-hypopnea index (AIH), specified in some studies and not in others. Second, the intervention included the continuous positive pressure in the airway. The treatment values were calculated in each case in an individualized polysomnography [11,12,13,14,15]. Third, different types of randomized, non-randomized, cohort, quasi-experimental and pilot studies were included. In some studies, a placebo CPAP was used consisting of masks with 10 holes of a quarter of an inch each [11,12,13,14]. In other cases the control was non-application or poor compliance [15]. Finally, the outcomes included were related to OSA and those that assess neurocognitive impairment. In the first group we found the results of the Epworth Sleep Scale (ESS) [11,12,14], the Respiratory Disturbance Index (RDI) [11,13], the Pittsburg Sleep Quality Index [14], the Sleep Functional Impact questionnaire [14] and the results of the polysomnography (PSG) [11,13]. To assess cognitive function, a battery of 10 neuropsychological tests was used in the case of Ancoli-Israel [12], the Cornell scale of depression in dementia together with a neuropsychiatric questionnaire [14] and finally the Mini Mental State Examination [12].

As inclusion criteria, it was proposed to analyze all the studies with CPAP intervention in patients with Alzheimer’s, whose languages were Spanish and English.

Those documents that were scientific comments and reviews of the state of the matter were excluded. Due to the specificity of the subject and the lack of related scientific production, the date of publication in the searches was not limited.

### 2.4. Evaluation of Clinical Relevance

The grades of recommendation have been studied according to the Duodecim (Finnish Medical Society Duodecim), a clinical practice guide developed in Finland to improve the quality of health care [17]. First, grade “A” means that the recommendation is based on strong evidence. Second, grade “B” is based on sufficient evidence to make a clear recommendation. Grade “C” recommendations are based on limited evidence. Finally, grade “D” refers to recommendations for which there is no evidence based on clinical studies [18,19].

Two independent reviewers (investigators Perez-Cabezas and Martín-Valero) completed the assessment list based on the PEDro score. The methodological quality and risk of bias were evaluated using the PEDro scale. This scale (0 to 10) is based on the list developed by Verhagen et al. [20], and assesses the internal validity of randomized controlled trials. A study with a PEDro score of 6 or more is considered level-1 evidence (6–8: good; 9–10: excellent) and a score of 5 or less is considered level-2 evidence (4–5: fair; <4: poor) [21].

## 3. Results

The literature search was conducted in January 2018 and updated in November 2019; 96 studies were obtained from all the databases. Duplicate results were eliminated. The flow chart representing the selection process is shown in Figure 1. Five documents were analyzed: three were clinical trials and two pilot studies, all published between 2006 and 2014. The main findings of this review are presented in Table 1. Table 2 shows an evaluation of methodological quality of the studies selected according to the PEDro scale.

The sample size in RCTs ranged from 39 to 52 patients. In the pilot study this was reduced to 10 and 28 subjects in each.

In four of reviewed investigations the intervention time is the same, 6 weeks total; during the first 3 weeks therapeutic CPAP was applied in the experimental group and CPAP placebo in the control. The rest of time all patients received therapeutic CPAP. Cooke et al. [14] designed a follow-up for 13 months after the end of the examination. Troussière et al. [15] analyzed the impact of CPAP on Alzheimer’s patients who had completed treatment for at least 3 months.

The most important results of the studies analyzed in this review are described below.

Chong et al. [11] demonstrated the efficacy of CPAP in patients with Alzheimer’s, considering variables related to OSA, such as the results in ESS and RDI.

Ancoli-Israel et al. [12] observed that although changes in neuropsychological functioning between treatment groups (comparing 3 weeks of tCPAP with 3 weeks of pCPAP) were not statistically different when considered individually, composite neuropsychological outcomes combined suggested modest and statistically significant improvements associated with cognitive functioning after 3 weeks of therapeutic CPAP. The composite score was defined as the mean of 14 standardized subscale scores on each of the neuropsychological battery subscales.

Cooke et al. [13] in their trial determined that after one night of treatment the tCPAP group showed statistically significant results compared to the placebo group at stage 1 (*p* = 0.04) and at sleep stage 2 (*p* = 0.02). In the paired analysis, the 3 weeks of tCPAP resulted in significant decreases in awake time after sleep (*p* = 0.005), in stage 1 (*p* = 0.001), in stage 3 (*p* = 0.006), in number of awakenings (*p* = 0.005).

The same author [14], in his pilot study, observed that continued use of CPAP in patients with Alzheimer’s may be associated with sustained benefits in sleep, mood and cognitive functioning. Subjects who continued with CPAP remained stable or showed improvement in measured values. However, those who did not continue with CPAP continued to deteriorate.

In 2014, Troussière et al. [15] found that the mean annual decrease in the Mini Mental State Examination was significantly slower in those patients with Alzheimer’s sleeping with the CPAP device. This suggests the treatment may slow cognitive deterioration.

The methodological quality and the degree of recommendation of the three clinical trials were evaluated. An RCT [11] marks the grade of recommendation A. The others [12,13] obtained grade B. The methodological quality according to the PEDro scale is shown in Table 2.

## 4. Discussion

This systematic review summarizes the levels of evidence and degrees of recommendation of therapeutic respiratory interventions for CPAP in AD patients. It has been observed that there are few experimental research works relating treatment to CPAP and its impact on AD. Nevertheless, we find an interest in the scientific field related to the topic, in which the presence of OSA and its relation with the cognitive deterioration in AD is questioned. These issues are discussed below.

The articles included in this review had a PEDro score of between four and eight, as shown in Table 2. Trials were considered of sufficient methodological quality if they had a score of at least 5 out of 10 points. This was based on the fact that the tests with a score close to 4 do not employ a triple blind methodology (i.e., patient, evaluator and providing treatment). Due to the type of treatment it is very difficult to obtain a triple blind; the placebo CPAP cannot go unnoticed by the therapist. However, the evaluator could have been blinded. This could not be possible in the trials consulted, since the evaluator and the therapist were the same person. In future studies it is recommended to include an evaluator other than the therapist, to increase the methodological quality.

Only one primary document [11] gave an “A” recommendation grade to treatment with CPAP in AD. This study shows strong evidence in favor of the application of CPAP in patients with AD; the results are related only to OSA variables (Epworth Sleepiness Scale and Respiratory Disturbance Index). The other studies consulted included evaluations of cognitive function. In this case the evidence is moderate.

Due to the above, on the one hand the impact of CPAP on OSA of Alzheimer’s patients is discussed and on the other hand the impact of CPAP on their cognitive impairment.

### 4.1. Alzheimer’s Disease and Obstructive Sleep Apnea

Sleep disorders are very common in diseases related to dementia, in fact it is estimated that 40% of AD patients present such disorders [22].

In 3 studies analyzed in this review, variables related to sleep quality and excessive daytime sleepiness are measured [11,13,14]. In all of them the results were in favor of CPAP [13].

One of the standard instruments in sleep analysis is polysomnography. It is a diagnosis and evaluation of the quality of sleep. It also serves to adjust the ventilation pressures for each of the patients, and is a personalized method [23]. For instance, all studies of the review include this method of diagnosis and evaluation [11,12,13,14,15].

The Epworth Somnolence Scale is included in two studies to see the daytime sleepiness of patients [11,14]. The excessive daytime sleepiness is one of the most common symptoms in OSA, as far as this outcome is concerned in all investigations related to the subject [24,25]. This instrument is presented as reliable in AD [22], so it should be included in future studies.

In AD patients it is very important to reduce the daytime sleepiness and the lethargy in which many of them are found. Even the caregivers consider that after the integration of the CPAP treatment the affected ones show less deterioration; and therefore the caregivers mood and quality of the sleep improved [14]. Due to the social implications of AD, the role of caregivers should be considered. AD patients require constant attention and for a very long period of time. Therefore, whether CPAP improves the perception of caregivers is another factor to take into account for its possible application.

### 4.2. Alzheimer’s Disease and Neurocognitive Impairment

Two studies included in this review assess neurocognitive function in AD patients after CPAP treatment. Ancoli-Israel et al. [12] measured this with a series of 10 neuropsychological tests, finding modest improvement in the therapeutic CPAP group at 6 weeks post-treatment. The results at 3 weeks were not significant so this period is not enough to obtain improvement. Troussière et al. [15] concluded that AD patients slow or suffer less cognitive impairment if treated with CPAP. This may be due to the lesions caused by OSA in nerve tissue. These statements coincide with the review of the topic by Ferini-Strambi [26], which states that OSA can accelerate the occurrence of mild cognitive impairment and Alzheimer’s disease and may also represent an independent risk factor for Parkinson’s disease. It is even mentioned that continuous positive airway pressure could delay the progression of AD, considering it an option of potential importance.

In relation to this, we can say that in elderly patients with OSA, cognitive deterioration and functional brain alterations are significantly higher [27]. A possible physiological explanation may be that oxidative stress, a process that is closely related to the formation and development of diseases of the nervous system, among them Alzheimer’s disease, is one of the most damaging consequences in the nervous tissue caused by OSA. Repeated airway obstruction and sleep collapse of OSA patients lead to chronic intermittent nocturnal hypoxia affecting mitochondria and causing dysfunction of the endoplasmic reticulum. For this reason antioxidant capacity is reduced, triggering DNA peroxidation damage and inflammatory response in the cerebral cortex and hippocampus. These changes could lead to apoptosis and necrosis of nerve cells and then contribute to neuropsychological disruption [26,28].

In the review, we have found studies whose participants were healthy elderly people with severe OSA in which it has been shown that after CPAP treatment their cognitive difficulties improve [29,30]. Even in the diagnosis by imaging it is observed that the cortical thinning is attenuated and the connectivity in the neural network increases [27].

Cognitive dysfunction caused by OSA in healthy elderly may have greater consequences in the Alzheimer’s population. For this reason, we emphasize the importance of diagnosing and treating OSA, especially in this type of patient, due to their vulnerability.

### 4.3. Limitations of the Study and Prospective Research

The review indicates the need to address this issue in the scientific field, due to AD’s relevance. The therapeutic use of CPAP is presented as a non-invasive and harmless option that may lead to a slow deterioration of this disease. However, very few experimental studies have been done so far. In addition, the disease stage of the patients is not specified.

Future investigations should include neuroimaging diagnostic tests in order to be able to check for possible changes in the cerebral cortex and frontal lobe, structures closely related to cognitive impairment in the elderly healthy [27]. In the sample inclusion criteria, the genetic load of the disease should be considered, and the possible mutation of three genes (*APP, PSEN1* and *PSEN2*) coding for the decomposition process of the amyloid precursor protein (APP) and of the generation of beta amyloid (Aβ). The mutation of these in the early presentation of the disease has been detected in more than 85% of those affected. For this reason they are considered diagnostic biomarkers in AD [5]. The mode of presentation of the disease is much more aggressive in this type of patient, so its peculiarities should be considered either excluding them from the sample or a different group.

For this reason, we propose future clinical trials with the following considerations:–The sample should be well delimited in terms of the evolution phase of AD, since the initial phase is the best time to see the CPAP impact on cognitive impairment [15].–Patients with AD that have a genetic load favorable to the disease should be excluded or considered in separate groups, due to the physiological implications of the genetic mutation in the oxidative process [31].–Neuroimaging tests should be included to assess possible brain changes [32].–It is very important to follow these patients to evaluate short- and long-term consequences.

## 5. Conclusions

CPAP treatment in AD patients decreases excessive daytime sleepiness and improves sleep quality. There are indications that cognitive function deterioration, measured with the Mini Mental Scale, decreases or evolves to a lesser extent in Alzheimer’s patients treated with CPAP. Caregivers observe stabilization in disease progression with integration of CPAP. More research is needed on the topic presented.

## Figures and Tables

**Figure 1 jcm-09-00181-f001:**
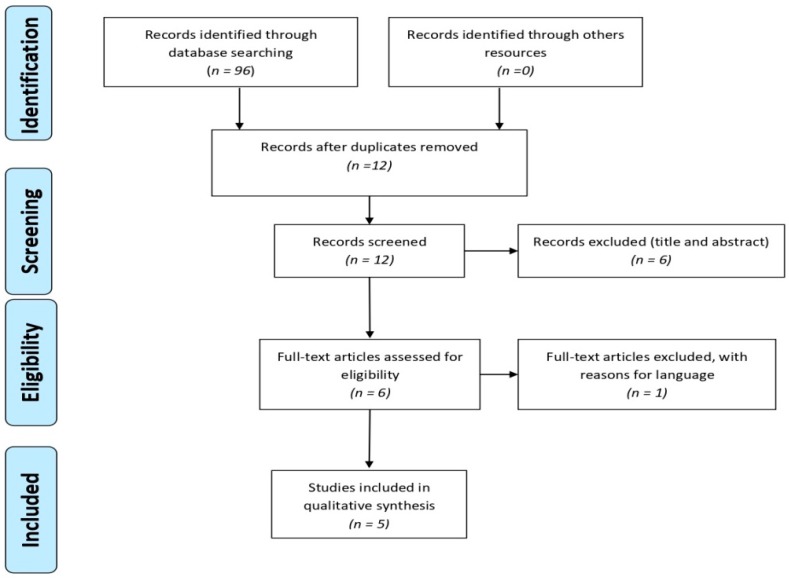
PRISMA flowchart [16].

**Table 1 jcm-09-00181-t001:** Main characteristics of the studies included in the review.

Study Type of Evidence	Age, Mean (Standard Deviation)	Sample Size (Complete the Study)	Variables	DR	Results
**Chong et al. 2006 [11]** **RCT (Randomized controlled trial)**	t-CPAP 77.7 p-group 77.9	39 t-CPAP group = 19 p-CPAP group = 20	ESS RDI	A	EES t-CPAP: 3 weeks *p* = 0.04 6 weeks *p* = 0.004 p-CPAP 3 weeks *p* = 0.33 6 weeks *p* = 0.06 RDI t-CPAP: 3 weeks *p* = 0.001 6 weeks *p* = 0.001 p-CPAP 3 weeks *p* > 0.05 6 weeks *p* = 0.001
**Ancoli-Israel et al. 2008 [12]** **RCT**	t-CPAP 78.6 (6.8) p-CPAP 77.7 (7.7)	52 (12 loss) t-CPAP group = 27 p-CPAP group = 25	14 Neuropsychol. tests (composite score) AHI	B	3 weeks NO significant differences. 6 weeks: *p* = 0.01 composite score
**Cooke et al. 2009 [14]** **Pilot study**	75.7 (5.9)	10 patients +9 caregivers +CPAP = 5 −CPAP = 5	PSQI ESS FISQ Cornell Scale Neuropsychology Inventory	-	—CPAP +: remained stable or improved in almost all measures —CPAP-continued to deteriorate —The caregivers of CPAP + patients improved their own quality of sleep, mood and reported patients stabilization.
**Cooke et al. 2009 [13]** **RCT**	t-CPAP 78.6 (6.8) p-CPAP 77.7 (7.7)	52 39 compled the study t-CPAP = 27 p-CPAP = 25	Polysomnography (1st night and at 3 weeks)	B	1st night T-CPAP Step 1 (*p* = 0.04) Stage 2 of sleep (*p* = 0.02) 3-weeks t-CPAP Time awake after sleep (*p* = 0.005) Step 1 (*p* = 0.001) Step 3 (*p* = 0.006). Awakenings (*p* = 0.005)
**Troussière et al. 2014 [15]** **Pilot study**	t-CPAP group 73.4 no-CPAP group 77.6	28 (5 loss) t-CPAP = 14 no-CPAP = 9	Mini Mental State Examination (MMSE)	-	T-CPAP = (−0.7) No-CPAP = (−2.2) MMSE *p* = 0.013

t-CPAP: Therapeutic continuous positive airway pressure (CPAP) group. p-CPAP: placebo CPAP group. ESS: Epworth Sleepiness Scale. RDI: Respiratory Disturbance Index. AHI: apnea-hypopnea index. DR: degrees of recommendation. PSQI. Pittsburgh Sleep Quality Index. FOSQ: Functional Outcome of Sleep Questionnaire. Grade “A” means that the recommendation is based on strong evidence. Second, grade “B” is based on sufficient evidence to make a clear recommendation.

**Table 2 jcm-09-00181-t002:** PEDro Score for Methodological Quality Assessment of 3 Studies.

Section Topic	Study		
	Chong et al. [11]	Ancoli-Israel et al. [12]	Cooke et al. [13]
1. Eligibility criteria were specified	Yes	Yes	Yes
2. Randomly allocated to groups	Yes	Yes	Yes
3. Concealed allocation	No	No	Yes
4. Comparability of base	Yes	Yes	Yes
5. Blinding of subjects	No	No	Yes
6. Blinding of therapists	No	No	No
7. Blinding of assessors	No	No	No
8. Proper continuation	No	Yes	Yes
9. Intention to treat	Yes	No	Yes
10. Between-group statistical comparisons	Yes	Yes	Yes
11. Point measure and measures of variability	No	Yes	Yes
Total	4/10	5/10	8/10

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
