# Peer review of "Continuous Positive Airway Pressure Treatment in Patients with Alzheimer’s Disease: A Systematic Review"

_jcm, 2020, doi:10.3390/jcm9010181_

Round 1
Reviewer 1 Report
Thank you for the opportunity to review this paper which addresses an important topic. Unfortunately I am unable to recommend this paper for publication in its current form.
Major concerns:
-the question is not well defined. The original question in prospero clearly stated the review was about neurocognitive improvements from CPAP treatment. Why not stick with this aim? Even if the conclusion states that only 1-2 studies were found. The new aim is very nebulous and confusing.
-The introduction does not reference accurate and current literature. For example the definition of OSA is incorrect. There are many other more credible sources from which to get an accurate definition of OSA. The statement that OSA can 'trigger' stroke and brain tumours is incorrect. There is some (conflicting) evidence from large cohort studies to suggest an association between OSA and stroke/cardiovascular disease. There is also conflicting evidence about the effect of CPAP on cognitive function with some studies suggesting no effect (eg APPLES) and others suggesting mild improvement in some domains.
-There is a repeated sentence in the introduction. I encourage the authors to go back to the literature to complete a more thorough review.
-description of the search strategy is poor and unnecessary detail is provided. Authors should keep this succinct and able to be replicated. The are many examples in the literature that could be followed.
- the correct name of the "Functional Impact of Sleep Questionnaire" (FISQ) is "Functional Outcome of Sleep Questionnaire" (FOSQ)
-Section 3 - Results. Whilst a critical appraisal checklist was performed, the authors have not taken these findings into account when describing and interpreting the studies, and therefore many of the conclusions are overstated. For example, the Ancoli-Israel study did not find any significant differences on any neuropsych measures, but a "modest" improvement when a composite score was developed. Is this composite score valid? Was the statistically significant difference also likely to be clinically significant? Line 168- Troussiere - a small uncontrolled pilot study with pre-post measurement cannot "affirm that the treatment stops cognitive deterioration".
I think the most interesting finding from this review is that there has only been one RCT that has looked at the effect of CPAP on neuropsych function in dementia.
I do believe this study could be published with a major revision. It is an important question. I encourage the authors to start with a more clearly defined question, to conduct a more thorough literature review (for the introduction and discussion), to be more critical with their interpretation of the studies, and to seek some assistance with the write-up and overall presentation of the manuscript.
Reviewer 2 Report
Thank you for the opportunity to review this manuscript. Previous studies have suggested a pathophysiological link between obstructive sleep apnea syndrome (OSAS) and Alzheimer's disease (AD). The aim of this study is to evaluate the degree of recommendation, considering the scientific literature, of continuous positive airway pressure (CPAP) treatment in AD patients. This article is very interesting. However, some items need to be modified or further clarified.
Introduction
Line 33: delete this sentence: “These pauses occur 30 times or more during sleep”. This is the definition of severe OSA.
Line 35-37: This sentence should be reviewed: Although the individual returns to normal breathing after these pauses, the accumulation of these can trigger atrial fibrillation, stroke, favor the onset of brain tumor and other vascular diseases causing death. It would be more correct to rewrite it this way: Obstructive Sleep Apnea represents a very important risk factor for stroke, hypertension, cardiovascular disease, hyperlipidemia, atrial fibrillation, type 2 diabetes mellitus and promote mechanisms that can increase a risk of cancer.
Discussion
Line 216: This sentence should be reviewed: “According to the literature reviewed, excessive daytime sleepiness is one of the most important symptoms in OSA”…
Reviewer 3 Report
The abstract has a spelling mistake. Introduction section: In what duration do 30 pauses need to occur? The definition of apnea is not clearly defined.
It's a well written article. Some corrections to the article are as follows:
1) Abstract: it has spelling mistake which needs to be corrected- continuous. 2) Introduction: the authors need to clearly defined obstructive sleep apnea. In their article, they mentioned pauses for 30 times or more. This is unclear. I believe the author is referring to severe obstructive sleep apnea where the AHI is 30 or more. They should also define the duration of event rather than mentioning some seconds to minutes. i.e. more than 10 seconds is an apnea or hypopnea.
Round 2
Reviewer 1 Report
Thank you for addressing many of the points raised in the initial review. I do believe the manuscript is now much more credible.
However, please consider changing the sentence:
"Thus, they affirmed that this treatment stops the cognitive deterioration." in Line 240 to something like "This suggests the treatment may slow cognitive deterioration."
I also think the manuscript needs a conclusion at the end, summarizing the overall findings.
Some further attention to English language would improve readability.
Author Response
Please see the attachment,
